# Isolation and Characterization of Extracellular Vesicles Secreted In Vitro by Porcine Microbiota

**DOI:** 10.3390/microorganisms8070983

**Published:** 2020-06-30

**Authors:** Leidy Lagos, Sabina Leanti La Rosa, Magnus Ø. Arntzen, Ragnhild Ånestad, Nicolas Terrapon, John Christian Gaby, Bjørge Westereng

**Affiliations:** 1Department of Animal and Aquaculture Sciences, Faculty of Biosciences, Norwegian University of Life Sciences, 1433 Aas, Norway; ragnhild.martinsen.anestad@nmbu.no; 2Faculty of Chemistry, Biotechnology and Food Sciences, Norwegian University of Life Sciences, 1433 Aas, Norway; sabina.leantilarosa@nmbu.no (S.L.L.R.); magnus.arntzen@nmbu.no (M.Ø.A.); jchristiangaby@gmail.com (J.C.G.); bjorge.westereng@nmbu.no (B.W.); 3Architecture et Fonction des Macromolécules Biologiques, UMR7257 CNRS AMU, USC1408 INRAE, 13288 Marseille, France; nicolas.terrapon@afmb.univ-mrs.fr

**Keywords:** extracellular vesicles, beta-mannan, microbiota, proteome, in vitro

## Abstract

The secretion of extracellular vesicles, EVs, is a common process in both prokaryotic and eukaryotic cells for intercellular communication, survival, and pathogenesis. Previous studies have illustrated the presence of EVs in supernatants from pure cultures of bacteria, including Gram-positive and Gram-negative glycan-degrading gut commensals. However, the isolation and characterization of EVs secreted by a complex microbial community have not been clearly reported. In a recent paper, we showed that wood-derived, complex β-mannan, which shares a structural similarity with conventional dietary fibers, can be used to modulate the porcine gut microbiota composition and activity. In this paper, we investigated the production, size, composition, and proteome of EVs secreted by pig fecal microbiota after 24 h enrichment on complex β-mannan. Using transmission electron microscopy and nanoparticle tracking analysis, we identified EVs with an average size of 165 nm. We utilized mass spectrometry-based metaproteomic profiling of EV proteins against a database of 355 metagenome-assembled genomes (MAGs) from the porcine colon and thereby identified 303 proteins. For EVs isolated from the culture grown on β-mannan, most proteins mapped to two MAGs, MAG53 and MAG272, belonging to the orders *Clostridiales* and *Bacilli*, respectively. Furthermore, the MAG with the third-most-detected protein was MAG 343, belonging to the order *Enterobacteriales*. The most abundant proteins detected in the β-mannan EVs proteome were involved in translation, energy production, amino acid, and carbohydrate transport, as well as metabolism. Overall, this proof-of-concept study demonstrates the successful isolation of EVs released from a complex microbial community; furthermore, the protein content of the EVs reflects the response of specific microbes to the available carbohydrate source.

## 1. Introduction

The distal gut microbiota has a profound influence on host health through the degradation of dietary fibers, production of metabolites and vitamins, maturation of the host immune system, and protection against pathogen invasion [1]. The gut ecosystem is characterized by a dynamic interaction between the microbiota and the host [2]. However, the gut microbiota does not directly interact with the intestinal epithelium. Commensal microbiota resides in the outer mucus layer, whereas the inner mucin layer is highly compacted and prevents bacteria from accessing the epithelial cells [3,4]. Nonetheless, an increase in the translocation of some gut microbial components into systemic circulation has been implicated in inflammation and cardiovascular events [5]. Still, the mechanism by which an interaction becomes established between a host and its intestinal microbiota, and also between different microbes, has been only partially addressed. Proteins and other metabolites secreted by the microbiota play an essential role in communication with intestinal cells [3,6]. Some of these metabolites are linked to environmental factors such as diet and lifestyle, and these are, therefore, described as diet-dependent metabolites [6]. The gut microbiota also generates various diet-independent metabolites, which include lipopolysaccharide (LPS) and peptidoglycan (PG) as well as extracellular vesicles [7]. The production and release of membrane vesicles is a process that is conserved across the three domains of life, Archaea, Bacteria, and Eukaryote [8]. Among Bacteria, cell envelope-derived EVs (also known as outer membrane vesicles or OMVs) from Gram-negative microbes have been reported and studied for decades [9]. Only recently, it has been shown that EVs production can occur in microorganisms with thick cell walls such as Gram-positive bacteria, mycobacteria, and fungi [10]. EVs range in size from 20 to 400 nm and play important roles in intercellular communication and signaling. They can also transport a variety of biomolecules such as enzymes, toxins, antigenic determinants, nucleic acids [11,12], and metabolites [13], which are protected from the extracellular environment of the gastrointestinal tract and from enzymatic degradation by the vesicle’s lipid bilayer envelope. EVs play essential biological roles not only in bacterial survival but also in host interactions, whereby they act as inter-kingdom signaling structures [14]. The secretion of EVs by Gram-negative bacteria such as *Bacteroides fragilis* is well known [13], while the release of EVs by Gram-positive phyla such as the *Firmicutes* and *Actinobacteria* due to medical and clinical implications have attracted more attention recently [15]. For instance, *Mycobacterium tuberculosis-*derived EVs that contain iron-binding factors contribute to bacterial survival, thus promoting infection, while *Staphylococcus aureus*-derived EVs that contain β-lactamase have a role in the spread of antibiotic resistance [16,17]. Nevertheless, most of the studies have focused on the secretion of EVs by pure cultures of bacteria, but not on the secretion of EVs by complex microbial communities like the ones found in the intestinal tract of animals.

Herein, we describe a proof-of-concept study to assess the feasibility of isolating EVs produced by a complex bacterial community in vitro. Before EVs isolation, the community derived from pig fecal samples was enriched using spruce (*Picea abies*) β-mannan. Furthermore, we show that the proteome of EVs from these intestinal bacteria is modulated by the available carbohydrate source.

## 2. Materials and Methods

### 2.1. Substrates Used

Acetylated galactoglucomannan (hereafter referred to as β-mannan) was produced in-house from steam-exploded Norway spruce, as described previously [18]. Mannopentaose, xylopentaose, cellopentaose, konjac glucomannan, carob galactomannan, and arabinogalactan from larch wood were purchased from Megazyme. Alpha-Mannan from *Saccharomyces cerevisiae* was purchased from Sigma. Acetylated arabinoglucuronoxylan was produced in-house following a protocol described in [19]. Glycan stocks (10 mg/mL) were prepared using purified H_2_O and sterilized by filtration using a 0.22-μm syringe filter (Sarstedt AG & Co, Nümbrecht, Germany).

### 2.2. In Vitro Fermentation

We evaluated the ability of β-mannan to promote the growth of extracellular vesicle-producing bacteria by using an in vitro, static batch culture fermentation system inoculated with fresh pig fecal inocula. To obtain anaerobic conditions, a 1 L conical flask with 800 mL of sterile basal medium (peptone 2 g/L, yeast extract 2 g/L, NaCl 0.1 g/L, K_2_HPO_4_ 0.04 g/L, KH_2_PO_4_ 0.04 g/L, MgSO_4_·7H_2_O 0.01 g/L, CaCl_2_·6H_2_O 0.01 g/L, NaHCO_3_ 2 g/L, Tween 80,2 mL/L, hemin 0.05 g/L, vitamin K 10 μg/L, L-cysteine hydrochloride 0.5 g/L, and bile salts (sodium glycocholate and sodium taurocholate at 0.5 g/L each)) was left overnight in an anaerobic chamber (Don Whitley Scientific, England, UK) under an atmosphere containing 85% N_2_, 5% CO_2_, and 10% H_2_. β-mannan, kept overnight in anaerobic conditions, was added to a final concentration of 2% (*w*/*w*) just before adding the fecal slurry at 0.01% *v*/*v*. The negative control consisted of a flask with a basal nutrient medium, but no added carbohydrate source. Before starting the experiment, pH was adjusted to 6.2–6.4 and the temperature of the anaerobic cabinet was set at 39 °C to resemble conditions in the porcine large intestine [20], but no pH-controller was used during the fermentation.

### 2.3. Fecal Inoculum

Porcine fecal samples were collected directly from the rectum of four healthy, weaned pigs (with bodyweight ranging from 18 to 22 kg) from three different pens; anaerobiosis was maintained using an anaerobic jar (AnaeroJar, 2.5 L, Oxoid, Hampshire, England, UK) equipped with a gas pack system (AnaeroGen, Oxoid, Hampshire, England, UK). To prepare the inoculum, equal amounts of feces (1.5 g wet weight) from each animal were pooled together and then diluted 1:5 in pre-reduced (oxygen-free) phosphate-buffered saline (PBS) as described in [20]. The fecal slurry was filtered to remove undigested fibers; the absence of fibers was confirmed using light microscopy. The resulting fecal slurry has a microbial population representative of the large intestinal microbiota of pigs when cultivated in vitro, as shown previously [21]. The fermentation was carried out for 24 h, which is comparable to the average transit time for material in the large intestine of monogastric animals [22]. At 0, 2, 4, 6, 8, and 24 h, the samples were collected for determining growth and pH. Growth was assessed by measuring the absorbance at 600 nm (OD_600_).

### 2.4. Scanning Electron Microscopy (SEM)

After 24 h fermentation, cells from three samples per condition were pelleted and pre-fixed in 50 μL of formaldehyde (37% *v*/*v*) following incubation for 15 min at room temperature. The fixed cells were washed three times in PBS (pH 7.4) and dehydrated in a gradient series of ethanol (50, 70, 90%, 95% and 3 × 100% for 10 min), followed by critical point drying with liquid CO_2_. Samples were coated with gold–palladium (Au–Pd) and examined in a Zeiss Evo 50 EP scanning electron microscope (Carl Zeiss AG, Oberkochen, Germany).

### 2.5. DNA Extraction and Sequencing

DNA was extracted from samples taken at 6 and 24 h with a MagAttract PowerMicrobiome DNA/RNA Kit (Qiagen, Hilden, Germany), according to the manufacturer’s instructions, except for the bead beating step where we used a FastPrep-96 Homogenizer (MP Biomedicals LLC., Santa Ana, CA, USA) at maximum intensity for a total of 3 min in three pulses of 60 s with a 5 min cooling period between each pulse. The extracted nucleic acids were quantified with a Qubit Fluorimeter and the Qubit dsDNA BR Assay Kit (Thermo Fisher Scientific, Waltham, MA, USA) and stored at −80 °C. SSU rRNA sequences were amplified using the primers Pro341F (5′-CCT ACG GGN BGC ASC AG-3′) and Pro805R (5′-GAC TAC NVG GGT ATC TAA TCC-3′), to which the MiSeq adaptors were additionally incorporated on the 5′ ends. The 25 µL PCR reactions consisted of 12.5 µL of iProof High-Fidelity Master Mix (Biorad, Hercules, CA, USA), 0.5 µL of a primer mix wherein each primer was at a 10 µM concentration, 2 µL template DNA, and 10 µL H_2_O. The PCR thermal cycling began with a hot start step at 98 °C for 5 min and was followed by 35 cycles of a 98 °C denaturation step for 60 s, a 55 °C annealing step for 30 s, and a 72 °C extension step for 30 s, which was then followed by a final 7 min extension step at 72 °C. The PCR products were purified using AMPure XP beads (Beckman Coulter, Indianapolis, IN, USA) and indexed with the Nextera XT Index Kit v2 (Illumina, San Diego, CA, USA) according to the Illumina protocol for 16S metagenomic sequencing library preparation. Next, equal volumes from each indexing reaction were pooled together, quantified with a Qubit Fluorimeter (Thermo Fisher Scientific, Waltham, MA, USA), diluted, denatured, and mixed with 5% PhiX Control v. 3 (Illumina, San Diego, CA, USA) according to the aforementioned Illumina protocol. The denatured library was sequenced on the Illumina MiSeq platform using the MiSeq Reagent Kit v. 3 (600 cycles). Data were retrieved from the sequencing machine as FASTQ files of the sequences for each demultiplexed sample.

### 2.6. Sequence Processing

Quality filtering, merging, denoising, and chimera detection were conducted to yield an amplicon sequence variant (ASV) table by using the DADA2 [23] package in R (R Core Team, 2019). The taxonomic classification of the ASVs was performed using Silva database release 132 as the taxonomy reference [24]. Further exploration and visualization of the amplicon sequence data were accomplished using the Phyloseq package in R [25]. For data visualization, taxa were agglomerated at either the phylum, order, or genus level. Following agglomeration, the abundances of taxa present at <1% abundance were summed and reported as a separate group in the stacked bar plots. All sequencing reads were deposited at the NCBI sequence read archive under BioProject PRJNA630859.

### 2.7. Isolation of Extracellular Vesicles

EVs were isolated through a centrifugation protocol [26,27] (Figure 1A). Briefly, the bacterial cells were removed by centrifugation (10 min, 15,000× *g*, 4 °C) and the supernatant was then filtered sequentially through 0.45 and 0.22 μm pore filters to remove the remaining bacterial cells. Then, the supernatant was concentrated from 1 L down to 0.1 L using a VivaFlow 200 cartridge with a 10 kDa cutoff (Sartorious AG, Goettingen, Germany) and further down to 4 mL using a centrifugal concentrator with a 100 kDa cutoff (Pall Life Sciences, Ann Arbor, MI, USA). The retentate was ultracentrifuged at 114,000× *g* for 2 h using an SW50.1 rotor in an Optima L-80 XP ultracentrifuge (Beckman Coulter, Krefeld, Germany). Pellets were resuspended in 5 mL PBS and centrifuged again at 114,000× *g*. After washing, the pellets containing EVs were resuspended in 1 mL PBS followed by purification on an Optiprep (Sigma-Aldrich, St. Luis, MO, USA) density gradient with centrifugation at 100,000× *g* for 18 h. To obtain a discontinuous density gradient, Optiprep solution was diluted with homogenization buffer (0.25 M sucrose, 1 mM EDTA, 10 mM Tris-HCl, pH 7.4) to obtain 40% (1.215 g/mL), 20% (1.111 g/mL), 10% (1.058 g/mL), and 5% (1.037 g/mL) iodixanol solutions. Aliquots of 4 mL with decreasing density were loaded sequentially into a 16.8 mL ultracentrifuge tube. Twelve 1 mL fractions were collected from the top. The protein concentration in the different fractions was determined using the Bradford protein assay (Bio-Rad Laboratories, Hercules, CA, USA) with bovine serum albumin as the standard. The EVs’ purity, approximate size, and morphology were verified by nanoparticle tracking analysis with a Zetasizer Nano ZS (Malvern Instruments Ltd., Malvern, UK). Ultrapure water at 25 °C was used as a constant parameter for viscosity and refractive index. The particle size (i.e., intensity-based size distribution plots and the intensity weighted mean hydrodynamic diameter) were expressed as the z-average. Data were analyzed using Zetasizer Software (v. 6.20) to calculate the hydrodynamic diameters of the particles.

### 2.8. Transmission Electron Microscopy of EVs

EVs samples were subjected to negative staining for TEM analysis. Formvar- and carbon-coated copper grids were incubated on a drop of EV suspension for 5 min. The grids were then washed three times with PBS and the adherent EVs fixed with 1% glutaraldehyde (Sigma-Aldrich, Darmstadt, Germany) for 4 min. Next, the grids were washed three times with PBS, two times with Milli-Q (MQ) water, stained for 20 s with 4% uranyl acetate (Sigma-Aldrich, Darmstadt, Germany) in MQ water, washed once with MQ water, and finally incubated on a solution of 1.8% methyl-cellulose (Sigma-Aldrich, Darmstadt, Germany) and 0.4% uranyl acetate for 10 min on ice. The grids were then dried and viewed in a Philips CM200 transmission electron microscope. Images were taken using a Quemesa camera and iTEM software (both Olympus soft imaging solutions, Munster, Germany).

### 2.9. SDS-PAGE

A standard SDS-PAGE procedure was used [28]. Briefly, 20 μg of isolated EVs were loaded onto a 12% (*w/v*) SDS polyacrylamide gel. The proteins were separated through SDS-PAGE and stained with Coomassie Blue, and the image was acquired and evaluated using Gel doc^™^ XR+ with Image Lab^™^ software (Bio-Rad, Munich, Germany). Protein molecular weight standards were obtained from Bio-Rad.

### 2.10. Proteomics and Bioinformatics Analyses

Two biological replicates of EVs per treatment were diluted to 20 µg of total protein in PBS, and the pH was adjusted to 8 by adding ammonium bicarbonate (Sigma-Aldrich, Darmstadt, Germany). Subsequently, the proteins were digested with 1 μg trypsin (Promega, sequencing grade) overnight at 37 °C. The digestion was stopped by adding 5 μL 50% formic acid and the generated peptides were purified using a ZipTip C18 (Millipore, Billerica, MA, USA) according to the manufacturer’s instructions, cleaned with micro-SPE, and dried using a Speed Vac concentrator (Concentrator Plus, Eppendorf, Hamburg, Germany). The tryptic peptides were dissolved in 10 µL 0.1% formic acid/2% acetonitrile and a 5-µL aliquot (~10 µg) was analyzed using an Ultimate 3000 RSLCnano-UHPLC system connected to a Q Exactive mass spectrometer (Thermo Fisher Scientific, Bremen, Germany) equipped with a nanoelectrospray ion source and loading capacity of 3.3 µg. For liquid chromatography separation, an Acclaim PepMap 100 column (C18, 2 µm beads, 100 Å, 75 μm inner diameter, 50 cm length, Dionex, Sunnyvale, CA, USA) was used. The mass spectrometer was operated in the data-dependent mode to automatically switch between MS and MS/MS acquisition. Survey full scan MS spectra (from m/z 400 to 2000) were acquired with the resolution R = 70,000 at *m/z* 200, after accumulation to a target of 1 × 10^5^. The maximum allowed ion accumulation times were 60 ms. The proteomic analysis was performed by the Proteomic core facility of the University of Oslo, Norway. The acquired raw data were analyzed using MaxQuant [29] v.1.4.1.2. Proteins were quantified using the MaxLFQ (label-free quantification) algorithm [30]. The data were searched against a sample-specific database consisting of 602,947 proteins that were generated from 355 metagenome-assembled genomes (MAGs) [18] from pig colon digesta and against the genome of *Sus scrofa* (40.708 sequences). Peptide identifications were filtered to achieve a protein false discovery rate (FDR) of 1% using the target-decoy strategy. Only proteins detected in both biological replicates for at least one of the conditions were considered valid EVs proteins. The mass spectrometry proteomics data were deposited with the ProteomeXchange Consortium via the PRIDE [31] partner repository with the dataset identifier PXD017420. The circular heat map was generated using R with the package circle based on the quantitative proteomics data in Appendix A.

### 2.11. Activity Assays

Enzymatic assays were conducted by adding 2 μL of either EVs or culture supernatant to 100 μL of a 1 mg/mL solution of the glycan substrates described in Appendix A. The reactions were carried out overnight in 10 mM sodium phosphate, pH 5.8, at 37 °C and 700 rpm shaking. The products were analyzed using Matrix-Assisted Laser Desorption/Ionization Time-of-Flight Mass Spectrometry (MALDI-ToF MS). MALDI-ToF analyses were performed using an Ultraflex MALDI-ToF/ToF instrument (Bruker Daltonics, Hamburg, Germany) equipped with a 337 nm nitrogen laser. Sample droplets were prepared by applying 2 μL of the matrix (9% 2,5-dihydroxybenzoic acid, DHB, in 30% acetonitrile (*v*/*v*)) and 1 µL of the sample, which was mixed directly onto an MTP 384 ground steel target plate (Bruker Daltonics GmbH, Hamburg, Germany). Droplets were dried under a stream of warm air.

## 3. Results and Discussion

### 3.1. Characterization of Microbiota Cultivated In Vitro

Several studies have shown a beneficial effect of β-mannan as microbiota-directed fiber (MDF) in a variety of gut microbiomes [32,33,34]. Therefore, it is interesting to study the specific bacterial taxa involved in its degradation and to infer possible mechanisms that mediate the interaction between the microbiota and its host. In this study, we used stool samples of pigs as inoculum and β-mannan as a substrate to stimulate the secretion of EVs from intestinal microbiota involved in the degradation of this MDF. Using metaproteomics, we identified the proteins, and hence the potential microbial functions embodied in the EVs. To begin, we determined whether β-mannan induces the in vitro growth of microbes in the fecal inoculum. During the period from 2 to 24 h, we noted a gradual increase in OD_600_ of the cultures (Figure 1B), while pH decreased (Figure 1C). The change of pH reflects the degree of fermentation, as shown previously [34]. Scanning electron microscopy indicated that the fecal inoculum includes a variety of bacteria of different morphologies, including cocci, diplococci, bacilli, chains of bacilli, and flagellate bacilli (Appendix A). The taxonomy of the microbial community was confirmed by 16S amplicon sequencing (Figure 1D–F). After filtering, trimming, and removing chimeras, a total of 21.902 and 42.740 amplicon sequences variant (ASVs) were detected in the microbial communities at 24 h in the control and β-mannan samples, respectively. Recently, we have shown that β-mannan influences the structure and composition of pig intestinal microbiota in vivo [18]. In the present in vitro study, the microbial community underwent changes in the presence of β-mannan, and we identified several taxa whose abundance shifted due to the presence of this MDF. The dominant taxonomic phylum in the initial community was *Firmicutes* (83%), which increased to 90% abundance in the presence of β-mannan when compared to 63% observed in control at 24 h. *Proteobacteria,* on the other hand, were present at only 4% in β-mannan vs. 30% in the control culture at 24 h (Figure 1D). Some *Firmicutes* are known as beneficial bacteria and can process plant polysaccharides into short-chain fatty acids (SCFAs), while *Proteobacteria* play a role in carbon cycling [35]. In the presence of β-mannan after 24 h cultivation, the orders *Lactobacillales* (40% abundance) and *Selemonadales* (39%) were dominant, whereas, at 24 h in control, they were 5% and 29% abundant, respectively (Figure 1E). However, in the culture without β-mannan at 24 h, *Enterobacteriales,* together with *Selenomonadales* were the most abundant orders (39% abundance). *Enterobacteriales* represented 5% abundance in the presence of β-mannan at 24 h (Figure 1E). Recently, Arfken et al. [36] showed that the main families present in the lower gastrointestinal tract of piglets were *Prevotellaceae, Lachnospiraceae*, and *Ruminoccoccaceae*, which can all degrade complex carbohydrates. Our study showed that the main genera present in the culture without β-mannan at 24 h were *Escherichia/Shigella* and *Megasphaera*, with 29% and 19%, respectively (Figure 1F), while *Lachnospiraceae* represented 4.8%. On the other hand, culture with β-mannan at 24 h showed that the main genus were *Lactobacillus* and *Megasphaera,* with an abundance of 44.5% and 32.5%, respectively. Interestingly, *Escherichia/Shigella* represented 5% abundance in the presence of β-mannan versus the 29% observed in the control. The difference between our results and the results reported by Arfken et al. [36] could be explained by the drop in pH observed at 24 h under in vitro conditions, which favored the growth of members of the Gram-positive *Lactobacillaceae*. Indeed, a low pH is known to cause a change in the composition of the gut microbiota, with decreased abundance of Gram-negative Bacteroides species, and predominance of Gram-positive Firmicutes [37].

### 3.2. Isolation of EVs from Pig Microbiota Cultivated In Vitro

Next, we studied the capacity of commensal bacteria to secrete EVs in vitro, whereby EVs were detectable after 24 h cultivation. The density of bacterial EVs was analyzed using Optiprep density gradient ultracentrifugations and ranged between 1.08 and 1.20 g/mL. The results from SDS-PAGE demonstrated that while a minor amount of EVs were present in the control culture (Figure 2A), EVs were abundant in the 2% β-mannan culture and accumulated in the fractions with an Optiprep density between 1.12 and 1.16 g/mL (Appendix A). These results were confirmed by transmission electron microscopy and nanoparticle tracking analysis (NTA), showing that EVs size and morphology were typical for bacterial EVs (Figure 2C–F). Further, we observed differences in the size of EVs isolated from the β-mannan culture vs. no-carbohydrate control (Figure 2C–F). While EVs isolated from the control culture without carbohydrate had an average diameter of 105 nm with a second peak at 350 nm, EVs from the β-mannan culture were larger, with an average diameter of 165 nm and a second peak at 950 nm(Figure 2D,F), which possibly corresponds to EVs aggregates as observed by TEM.

### 3.3. Proteome of EVs

The protein content of EVs from the β-mannan and control samples was analyzed by mass spectrometry. The database used for matching during proteomics analysis has been previously generated by our group [18] and contains 602,947 proteins inferred from 355 MAGs. A total of 303 proteins were identified, of which 28% were detected on EVs isolated from both the β-mannan culture and the β-mannan-free, whereas 55% were unique proteins identified just in the EVs isolated from the β-mannan culture (Appendix A). Most of the proteins identified from the EVs isolated from the β-mannan samples mapped to MAG53 and MAG272, which belong to the phylum *Firmicutes*, from the orders *Clostridiales* and *Bacilli*, respectively (Appendix A, highlighted). Furthermore, the MAG with the third-most detected proteins was MAG343, which belongs to the phylum *Proteobacteria*, order *Enterobacteriales* (Figure 1 D–E). We found several proteins mediating the secretion of EVs such as outer membrane proteins OmpA [38] in MAGs 53, 272, and 343. To determine whether the isolated EVs may contain enzymes that are able to degrade carbohydrates, as described previously for EVs isolated from *Fibrobacter succinogenes* [26], we performed a carbohydrate degradation assay. The supernatant of the culture grown in the presence of β-mannan contained enzymes capable of degrading mannopentaose, konjac glucomannan, carob galactomannan, and acetylated galactoglucomannan, indicating the presence of secreted endo-β-mannanases (Appendix A). This is in line with previous studies showing that *Bifidobacteria*, which were abundant in the β-mannan enriched samples as detected by the 16S analysis, secrete β-mannanases into the growth medium to confer degradation of mannan to oligosaccharides that are subsequently internalized [39]. No enzymatic activity on other hemicelluloses and α-mannan was detected when using purified EVs (Appendix A), consistent with the fact that enzymes with predicted hemicelluloses- and α-mannan-degrading functions were not identified in the β-mannan EVs proteome.

Classification of the identified proteins present in EVs into Clusters of Orthologous Groups (COG) of proteins revealed that the most abundant proteins belonged to MAGs 53, 272, and 343, and are involved in translation, ribosomal structure, RNA metabolic processes, and nucleotide-binding (Figure 3). Interestingly, Zhang et al. [40] showed an association between microbiome and intestinal EVs in pediatric inflammatory bowel disease, where the proteins found in EVs isolated from microbiota were related to DNA replication, recombination, and repair. In addition, S. Liu et al. [41] found that miRNAs were present in EVs isolated from feces and that they directly regulated specific bacterial expression affecting microbial growth.

In our study, most of the identified proteins from the EVs were assigned to MAG 272, where they participated in energy production, metabolism, transport of lipids or amino acids, and carbohydrate metabolism (Figure 3). The same trend was observed in MAG53, phylotype *Clostridiales*, and MAG343, which corresponds to *Escherichia coli,* a classical, intestinal, Gram-negative commensal (Figure 3). As bacteria proliferated, the secretion of EVs increased, in line with the increase of the relative abundance of taxa found by 16S sequencing. However, according to the 16S results, the order *Selenomonadale*s was present at 39% in the β-mannan culture after 24 h fermentation, but the proteome profile of EVs detected only MAG317 (Appendix A) as belonging to that order. These results may have two explanations, either the order *Selenomonadales* secretes a very low number of EVs in our in vitro conditions or the proteome of the microbial community differ to the one obtained by 16S sequencing. The second explanation is supported by the study of Zhang et al. [40], where they performed a metagenomic analysis of human feces and determined the proteome of EVs isolated from the same feces. The study showed that the microbial community composition of EVs differs from that of the fecal samples as assessed by 16S amplicon sequencing. These results exemplify the use of proteomics as a tool to identify specific proteins related to EVs function.

Interestingly, among the proteins present in the EVs isolated from both cultures, we detected proteins belonging to the flagellin cluster. Considering that the density and size of pili structures and EVs are similar [42,43], together with the identification of these structures by TEM (Figure 2E, black arrows), the method of progressive microfiltrations and ultra-centrifugation used in this study isolates both EVs and pili from bacteria present in a complex microbial community.

Furthermore, some of the proteins found in the fractions containing β-mannan-induced EVs were associated with fiber-degrading *Prevotellaceae* (MAGs 34 and 346, Appendix A green highlights) with proteins annotated as being involved in glycan binding and catabolism. Interestingly, MAG34 contains 18 PULs (polysaccharide utilization loci) and MAG346 contains eight PULs. In MAG34, at least one PUL seems to be mannan-specific, containing 11 subcomponents, including the glycoside hydrolases (GH) GH130 and GH26, SusC, SusD, epimerase, acetyl esterase and an inner membrane symporter [44]. This suggests that MAG34 is indeed capable of degrading β-mannan and further corroborates previous in vivo experiments, where, for the most part, proteins originating from the fiber-degrading *Prevotella* were more abundant in β-mannan-fed pigs [18].

## 4. Conclusions

We succeeded in isolating and analyzing the protein content of EVs secreted from a complex microbial community. β-Mannan induced the secretion of EVs from *Clostridiales*, *Bacilli*, and *Enterobacteriales* (MAGs 53, 272, and 343); however, the EVs did not contain polysaccharide degrading enzymes, contrary to previous observations for *Fibrobacter* [26]. None of these three MAGs seems to be prominent polysaccharide (or mannan) degraders from the low repertoire of CAZymes in the genomes. Therefore, the link between β-mannan and increased EVs production remains unidentified. However, the composition and amount of microbial EVs seem to depend on the conditions of the environment, such as the available carbohydrate sources, which in this case, was β-mannan. The role of EVs as mediators of inter-species interactions and function in the gut warrants further inquiry.

## Figures and Tables

**Figure 1 microorganisms-08-00983-f001:**
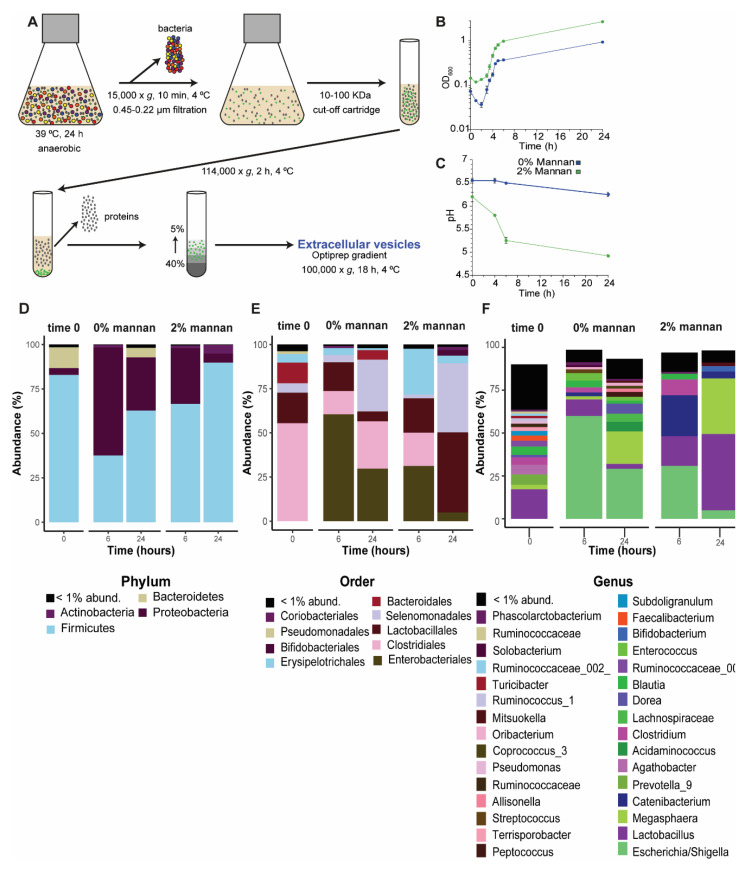
Methodology to isolate extracellular vesicles, EVs, and characterization of the bacterial community. Diagram showing the methodology used to isolate EVs in vitro, based on sequential centrifugation and density gradient separation (**A**). Growth curve of fecal inoculum cultivated in vitro in the presence of β-mannan or without any carbohydrate (control) (**B**). pH measurement of in vitro cultures either with or without the presence of β-mannan (**C**). Microbial diversity and composition in the cultures containing either 2% β-mannan or no carbohydrate addition. Samples were collected at 0, 6, and 24 h and subjected to amplicon sequencing of the 16S rRNA gene. The relative abundances of the dominant phyla (**D**), orders (**E**), and genera (**F**) that resulted from in vitro cultivation in the absence (0% mannan) or presence (2% mannan) of β-mannan are shown as stacked bar plots. The black bars in D–F indicate the summed abundance of different phyla, orders, or genera that were individually present at less than 1% abundance.

**Figure 2 microorganisms-08-00983-f002:**
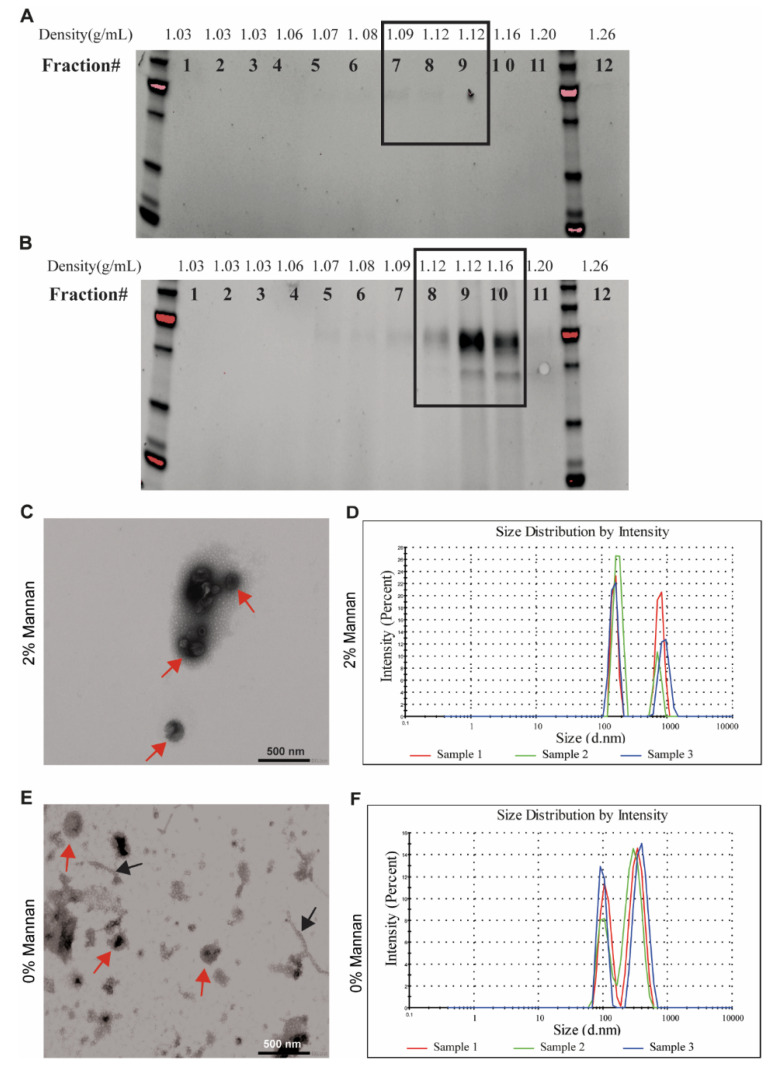
Characterization of secreted EVs from a pig fecal inoculum. The fecal inoculum was cultivated in vitro either in the presence of 2% β-mannan or without any carbohydrate source for 24 h before removing the cells from the culture medium. Density of EVs isolated from fecal culture on basal medium (**A**) or supplemented with 2% β-mannan (**B**) using a density gradient. Twelve fractions were recollected from the top of the gradient and analyzed using SDS-PAGE. The electron microphotograph shows that the sample recovered after ultracentrifugation and density gradient separation was bacteria-free and consisted of vesicles shaped as spheres in the presence of 2% β-mannan (**C**) or without carbohydrate (**E**). The red arrow indicates EVs, while black arrows indicate “pili-like” structures. The scale bar indicates 500 nm. Size of the vesicles isolated in the presence of 2% β-mannan (**D**) or without carbohydrate (**F**) was visualized by a Zetasizer uV.

**Figure 3 microorganisms-08-00983-f003:**
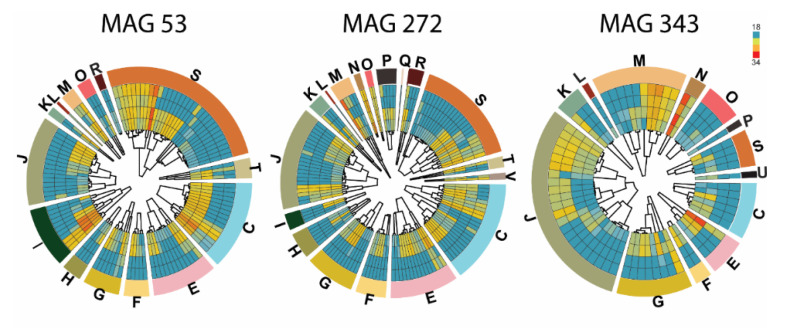
The proteome of EVs isolated from fecal inoculum cultivated in vitro for 24 h. The inner part of the circles displays a dendrogram and a heatmap of the identified proteins within EVs from MAG 53, 272, and 343 when grown with no carbohydrate (two outer rings) or with mannan (two inner rings). The colors in the heatmap indicate protein abundance and range from low abundance (blue; log2(LFQ) of 18 or lower) to high (red; log2(LFQ) of 34). The outer part of the circle shows Clusters of Orthologous Groups (COG) of proteins represented as one-letter-codes that are described below. Information storage and processing; **[J]** Translation, ribosome structure and biogenesis, **[A]** RNA processing and modification, **[K]** Transcription, **[L]** Replication, recombination and repair, **[B]** Chromatin structure and dynamics. Cellular processes and signaling; **[D]** Cell cycle control, cell division, chromosome partitioning, **[Y]** Nuclear structure, **[V]** Defense mechanisms, **[T]** Signal transduction mechanisms, **[M]** Cell wall/membrane/envelope biogenesis, **[N]** Cell motility, **[Z]** Cytoskeleton, **[W]** Extracellular structures, **[U]** Intracellular trafficking, secretion and vesicular transport, **[O]** Post-translational modification, protein turnover, chaperones. Metabolism; **[C]** Energy production and conversion, **[G]** Carbohydrate transport and metabolism, **[E]** Amino acid transport and metabolism, **[F]** Nucleotide transport and metabolism, **[H]** Coenzyme transport and metabolism, **[I]** Lipid transport and metabolism, **[P]** Inorganic ion transport and metabolism, **[Q]** Secondary metabolites biosynthesis, transport and catabolism. Some are poorly characterized in literature; **[R]** General function prediction only, **[S]** Function unknown.

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
