# Peer review of "Isolation and Characterization of Extracellular Vesicles Secreted In Vitro by Porcine Microbiota"

_microorganisms, 2020, doi:10.3390/microorganisms8070983_

Round 1
Reviewer 1 Report
The manuscript proposed by Lagos L et al. about the ‘Isolation and characterization of extracellular vesicles secreted in vitro by porcine microbiota’ provides valuable information about EVs released from microbial community, focusing on in vitro isolation and characterization of extracellular vesicles secreted.
In order to standardize the experimental approaches, studies based on the isolation and characterization of microvesicles should always refer to the lates MISEV directives (Théry C et al.; ‘Minimal information for studies of extracellular vesicles 2018 (MISEV2018): a position statement of the International Society for Extracellular Vesicles and update of the MISEV2014 guidelines’. J Extracell Vesicles. 2018. PMID: 30637094). It is important to underline that these studies, including the one presented here, will allow us to further expand the knowledge of extracellular vesicles released by microorganisms.
The manuscript shows preliminary results (exclusively isolation and characterizaion of extracellular vesicles from porcine microbiota), is overall well written and requires minor revisions.
I suggest to move Supplementary Figure S1C-D in the main text; furthermore, difference in proteins profile between extracellular vesicles and whole cell lysate should be showed through SDS-PAGE analysis by using Coomassie brilliant blue G250 dye.
Lipid A (LPS and LPA) are expressed in extracellular vesicles from microbiota (WB analysis and/or ELISA assay)?
Reviewer 2 Report
In the reviewed article, the Authors made “proof-of-concept study” and demonstrated obtained results but at the end of proof-reading I still don’t know what for. I hope that some explanations will be included in the improved version of the article.
Below are more itemized comments:
l.18 on the basis of what data the authors state: “ complex β-mannan, which shares structural similarity with conventional dietary fibers”; analysis of the basic composition of β-mannan (made by Authors in this study) after its hydrolysis and derivatization to alditol acetate and analysis with the GC-MS technique does not give such a response; similarly data in ref [18]
l.30 please edit this sentence; “whereby” should be replaced with other word
l.62-64 “While … clinical implication” - illogical sentence requires editing for example as below:
The secretion of EVs by Gram-negative bacteria such as Bacteroides fragilis is well known, while the release of EVs by Gram-positive phyla such as the Firmicutes and Actinobacteria due to medical and clinical implications have attracted more attention recently.
l.64-67 “These include … antibiotic resistance” - very complex sentence, which makes understanding difficult; in addition, grammatically incorrect – please edit it; keep the past tense as in the previous sentence and may divide it into two separate sentences which will make the statement more transparent
l.70-72 “Herein, … β-mannan.” as above, please edit it
l.86 instead “vessel” use “flask or conical flask” - herein and everywhere further in the text; instead
“A 1 L … H2” I would recommend “To obtain anaerobic conditions, 1L conical flask with 800 mL of sterile basal medium (medium composition in brackets) was left …. H2”
l.91 “Anaerobic β-mannan” – what does it mean?
l.92 instead “reach a” please use “final” ; And again edit it “Just before inoculation of culture/ or medium, ….”
l.93-96; “Temperature … the fermentation.” I would recommend “Before starting the experiment, pH was adjusted to 6.2-6.4 and temperature of the anaerobic cabinet was set at 39 °C to resemble conditions in the porcine large intestine.”
l.96-104; I would recommend to separate this part as a subsection (2.2) "The preparation of fecal inoculum"; additionally please omit “was weighed inside the anaerobic cabinet,”
l.102 What is “a tea sieve”? I would recommend to omit it and combine two sentences into one
“The fecal slurry was filtered to remove undigested fibers what was confirmed using light microscopy”
l.103 “was assumed” suggest that Authors are not sure, thus I would recommend to edit it
l.104 “Ten mL … this experiment.” Data from this sentence should be included in the sentence from l.92, and it is not necessary to write “all vessels used in this experiment.”
l.105 pleas omit “Once inoculated”
l.107 rather “At 0, 2, 4, 6, 8 and 24 h, samples were … to determine ..”
l.110 instead “Microscope” please use „ Microscopy”; moreover, the description of specimen preparation for SEM is not complete (the conditions of fixation, dehydratation, …, sputter coating with conductive material are not included); and not “resuspended” but “pre-fixed” etc.
l.158 I would recommend changing into “After washing, the pellets containing EVs were resuspended in 1 mL PBS followed purification on Optiprep (Sigma-Aldrich) density gradient with centrifugation at 100 000 g for 18 h.” then “EVs were resuspended in 1 mL PBS and were layered atop the 5% iodixanol solution and centrifuged at 100 000 g for 18 h” should be removed
l.161 if the Optiprep with the highest density is at the bottom of the tube then it should probably be "decreasing" instead of "increasing"
l.166 It seems to me that instead of “approximate purity, size, and morphology“ should be “purity, approximate size, and morphology”
l.184 TEM preparation description is not complete (e.g. dehydratation conditions are not included)
l.195 please add references for “standard SDS-PAGE procedure”
l.208 rather use “5-µl aliquots”
l.227 “Activity assays” description is not complete (e.g. final volume of reaction mixture); what enzymes activity was detected? Was the final concentration of EVs calculated as protein in the reaction mixtures the same?
l.239 pepton and yeast extract are good carbon sources for bacteria; I think that Authors ment “carbohydrate” not “carbon source”
l.247 I think that here should be plural form “β-mannans”
l.254 Shouldn’t be sth like “During period from 2 to 24 h, we noted a gradual increase in OD600 of the cultures, (Fig. 1B), while pH much decreased”
l.261 omit “both” since, it suggest the same results for control and tested conditions (word “respectively” excluds “both”)
l.272 Please check if there should be 19% instead of 29%, which would result from the next sentence
l.277-300 the past tense and passive voice should be kept all the time and not alternate with the present (examples: l.277 “shows” → “showed”, l. 279 “represent” and many more)
l.279 and 280 convert comma to dot in percentages
l.281 this is a repetition of information from l. 274 and l. 278; please clarify
l.282 “Our different results” for me this statement is incomprehensible
l.294 “EVs originated from live bacteria” I don’t understand how TEM can resolve this problem; I suggest changing this part to:
These results were confirmed by transmission electron microscopy and nanoparticle tracking analysis (NTA), showing that EV size and morphology were typical for bacterial EVs To confirm that the EVs originated from live bacteria, we used TEM to analyze the EVs secreted in the culture without carbohydrate; results showed differences in the size of EVs isolated from the β-mannan culture (Fig. 2A-D). vs. no-carbohydrate control (Fig. 2A, C).
These results were confirmed by transmission electron microscopy and nanoparticle tracking analysis (NTA), showing differences in the size of EVs isolated from the β-mannan culture vs. no-carbohydrate control.
l.324 omit second “and” at the end of the line
l.330 Before “Most of the identified …” please add “In our study”, since in this chapter results and discussion are plotted
l.336 “Selenomonadales was present at 30%” and in the l. 271 “Selemonadales (39 %)”; please correct
l.342 “The study shows …” sentence in the past tens
l.344 Maybe instead “contributing” it would be better use “related”
l.366 plili structures are not showed/indicated in Fig. 2C
l.378 I would recommend to make a separate chapter 'conclusions' from the last paragraph
According to “Instruction for Authors” References should be described as follows:
Journal Articles:
- Author 1, A.B.; Author 2, C.D. Title of the article. Abbreviated Journal NameYear, Volume, page range.
Please edit the signature of Figure 2S “ MALDI-TOF analysis of degradation products obtained from oligosaccharides as a result of action …. ; moreover the explanation of all abbreviations should be given at the end
In the signature of Figure 1S I think it should be "grown"
